# Preparation and Thermal Model of Tetradecane/Expanded Graphite and A Spiral Wavy Plate Cold Storage Tank

Hongguang Zhang [1,2,†], Tanghan Wu [2,†], Lei Tang [2], Ziye Ling [2,*], Zhengguo Zhang [2,*] and Xiaoming Fang [2]

1  Foshan Shunde Midea Electrical Heating Appliances Manufacturing Co., Ltd., Foshan 528311, China
2  Key Laboratory of Enhanced Heat Transfer and Energy Conservation, The Ministry of Education, School of Chemistry and Chemical Engineering, South China University of Technology, Guangzhou 510640, China
*  Correspondence: zyling@scut.edu.cn (Z.L.); cezhang@scut.edu.cn (Z.Z.)
†  These authors contributed equally to this work.

**Abstract:** A cold storage unit can store the cold energy off-peak and release it for building cooling on-peak, which can reduce the electricity load of air conditioning systems. n-tetradecane is a suitable cold storage material for air conditioning, with a phase change temperature of is 4–8 °C and a phase change enthalpy of 200 kJ/kg. However, its low thermal conductivity limits the application of n-tetradecane for high-power cold storage/release. This paper prepares a tetradecane/expanded graphite (EG) composite phase change material (CPCM), whose thermal conductivity can be increased up to 21.0 W/m·K, nearly 100 times over the raw n-tetradecane. A novel model to predict the maximum loading fraction of paraffin in the EG matrix is presented, with an error within 1.7%. We also develop a thermal conductivity model to predict the thermal conductivity of the CPCM precisely, with an error of less than 10%. In addition, an innovative spiral wave plate cold storage tank has been designed for the tetradecane/EG composite. The power and energy density of the cold storage tank are significantly improved compared to that of raw tetradecane. The energy density reaches 40 kWh/m³, which is high among the organic PCM thermal storage tank. This paper shows the significance of thermal conductivity enhancement in designing a cold storage tank.

**Keywords:** thermal model; cold storage; phase change materials; tetradecane/expanded graphite

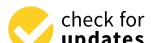

## 1. Introduction

A large market is emerging for cold storage, which is used for storing low-temperature thermal energy and releasing it for cooling. A wide range of applications for cold storage have been developed, such as cold chain logistics [1], air conditioning systems [2], etc. Phase change materials (PCMs) are a major kind of medium for cold storage that can absorb a large amount of heat during a solid-liquid phase change and maintain the temperature within a constant range. PCMs have the advantages of large thermal storage capacity and have become a commercial choice for cold storage.

Organic and inorganic PCMs are two major categories for cold storage materials. The organics mainly consist of paraffins such as dodecane [3], tetradecane [4], or eutectic of alphabetic acids [5], alcohols [6], or esters, such as the eutectic of dodecyl alcohol and octylic acid, palmitates, etc. The inorganics include ice [7], tetrabutyl-(ammonium/phosphonium) salts [8], and calcium chloride hexahydrate [9,10]. Among them, n-tetradecane has a melting point of 4 °C, which is suitable for a cold storage box in the transportation of cherished food or medical products as they need to be contained at a temperature between 2 and 8 °C. This melting point is also suited for the production of chilled water around 7–8 °C used in air conditioning. Moreover, the organic n-tetradecane is more chemically stable compared with most inorganic PCMs.

However, the thermal conductivity of the n-tetradecane, 0.2 W/m·K, falls outside the need for most cold storage applications. Cold storage boxes need the PCM thermal

conductivity to be lowered to insulate the heat from the surroundings. On the contrary, cold storage tanks usually require a much higher thermal conductivity for rapid cold storage and retrieval.

To design a fast-responding cold storage tank, this paper intends to enhance the thermal conductivity of n-tetradecane. Works in the literature have made attempts of adding thermally conductive fillers to enhance the conductivity of PCMs [11]. Expanded graphite (EG) has been found to be extremely effective at improving the thermal conductivity of the PCM. EG also has a porous structure to infiltrate PCMs, which allows the material to be shape-stabilized. The PCM/EG composite remains solid on macroscopic- no liquid leaks from the porous EG matrix even the PCM melts and the difficulty in container sealing is reduced. Therefore, we adopt EG to enhance the thermal conductivity of n-tetradecane. Thermophysical properties of the EG/PCM could be regulated with the density of EG. The increase in EG density gives the composite a higher thermal conductivity. On the other hand, a higher density may squeeze the pore volume of EG and lower the maximum loading percentage of PCMs. A model is needed to describe the effect of EG density on the thermal conductivity and enthalpy of the paraffin/EG composite.

Apart from the composite PCM with a high thermal conductivity, an efficient heat exchanger should be designed to match the heat transfer performance of the material. Shell and tube [12], plate to air [2], tube in plate [13], and tube and fin [14], and other types of cold storage units have been studied. However, the power density of the cold energy storage units above was limited due to the poor thermophysical properties of the PCMs or the heat transfer structure. A cold storage tank with a high thermal power density is needed for the cold energy demand in the air conditioning systems.

This paper presents a method to prepare n-tetradecane/EG composite PCM with a thermal conductivity above 20 W/m·K and develops a theoretical model to predict the material's thermophysical property. We also put forward with a novel heat exchanger with a spiral wave plate. This highly compact structure provides an efficient combination with the composite PCM with high thermal conductivity to store and release cold energy rapidly.

## 2. Experiment

### 2.1. Material Preparation

n-tetradecane (purity: 98.5%) was purchased from Guangzhou Zhongjia New Material Technology Co., Ltd. (Guangzhou, China). EG was purchased from Qingdao Furuite Graphite Co, Ltd. The EG was dried at 105 °C for 12 h in order to remove the adsorbed water on the surfaces.

EG was compressed into a cylinder of $\varphi 40 \times 10$ mm, with different densities of 50, 150, 250, and 350 kg/m$^3$. The EG cylinders were then immersed into the paraffin n-tetradecane. The n-tetradecane infiltrated into the EG pores to form the composite PCM. The mass changes during a 400 min paraffin soaking process were recorded every 50 min. The absorption rate of paraffin in the EG pores was also compared with different soaking temperatures from 15 to 55 °C.

### 2.2. Thermophysical Properties Characterization

The phase transition temperatures and latent heats of the pure phase change materials and CPCM were measured by a Q20 DSC differential scanning calorimeter from TA. The thermal conductivity of the PCMs were measured with a TPS 2500S Hot Disk thermal constant analyzer from Swedish Hot Disk Ltd. (Gothenburg, Sweden).

### 2.3. Cold Storage Unit Design and Numerical Simulation

As shown in Figure 1, a wave-like spiral plate cold storage tank was designed. Its energy density was supposed to be 200 kJ, with a discharge power of 0.5 kW. This cold storage tank is one-tenth unit of a cold storage system with a capacity of 0.5 kWh and 5 kW, which is used for emergency cooling for a small power system. On the base of a high-power density spiral plate heat exchanger, we introduced a wave-shaped structure

to the plate. The wavy structure was supposed to increase the turbulent disturbance of fluid flow and enhance the heat transfer. The thickness of the plate was 5 mm with a wall thickness of 1 mm. Between the neighboring layers of plate, the PCMs were filled with an initial thickness 10 mm. The inlet temperature of the warm water is fixed at 10 °C, and the flow rate is 200 L/h.

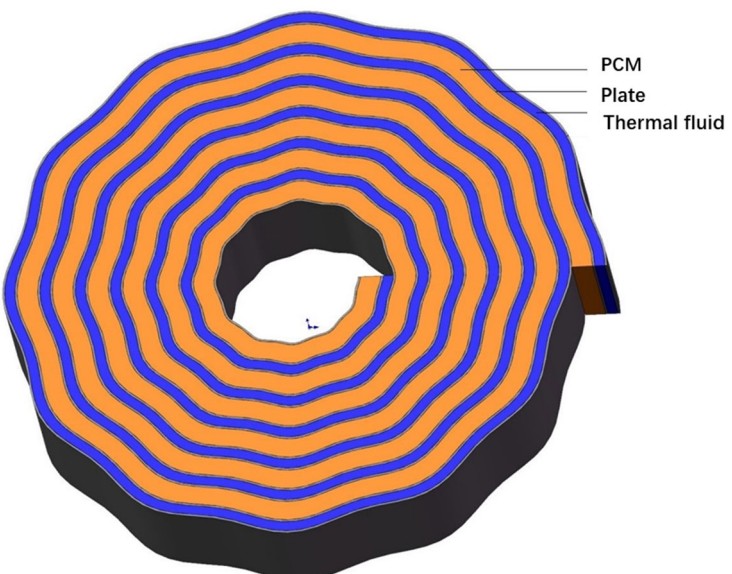

**Figure 1.** Structure of the wave-like spiral plate cold storage tank.

The wavy boundary of the spiral plates was governed by the following equations:

$$x = cos\theta \times \left(150 - 17 \times \frac{\theta}{2\pi}\right) + \left(3 - \frac{3}{150} \times 17 \times \frac{\theta}{2\pi}\right) \times cos12 \tag{1}$$

$$y = sin\theta \times \left(150 - 17 \times \frac{\theta}{2\pi}\right) + \left(3 - \frac{3}{150} \times 17 \times \frac{\theta}{2\pi}\right) \times cos12 \tag{2}$$

The heat transfer process was simulated numerically, with governing equations as described by He et al. [15]. The model had also been validated in this reference.

To optimize the structure of the spiral wavy plate cold storage tank, we must take a balance between the energy and power density. We fixed the energy storage capacity of PCMs but compared the performance of the tank that was filled with PCMs with different conductivities. If the PCM had a higher thermal conductivity, the heat transfer rate could be enhanced- and the plate volume could be smaller. However, its specific phase change enthalpy was lower, which required a larger volume of PCMs. This paper intends to find out sensitivity of the thermophysical properties to the energy and power density.

## 3. Results and Discussions

### 3.1. Shape Stability of n-Tetradecane/EG

Figure 2 shows that the appearance of the n-tetradecane/EG composite at a room temperature of 25 °C. The cylinder has a diameter of 40 mm, a thickness of 10 mm, and weighs 14.3 g. At room temperature, the n-tetradecane is actually in a liquid state. However, we cannot observe any liquid traces on the surface of the sample. The porous EG maintains the n-tetradecane/EG composite in a stabilized shape. No liquid leaks out from the composite PCM, even when the tetradecane melts.

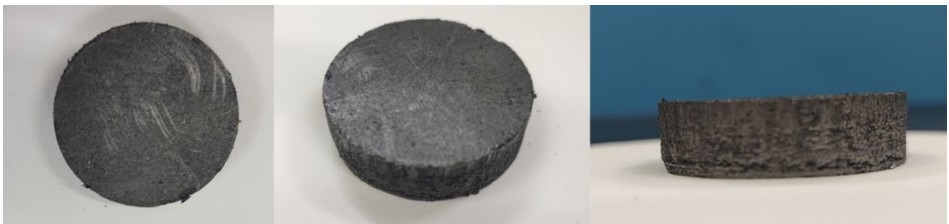

**Figure 2.** Photos of n-tetradecane/EG composite.

Figure 3 illustrates how the n-tetradecane infiltrates the EG matrix. The capillary force drives the liquid to infiltrate into the micropores. The liquid paraffin is confined in the micro spaces of the EG and is not likely to leak out. As a consequence, the composite PCM is solid on macroscopic.

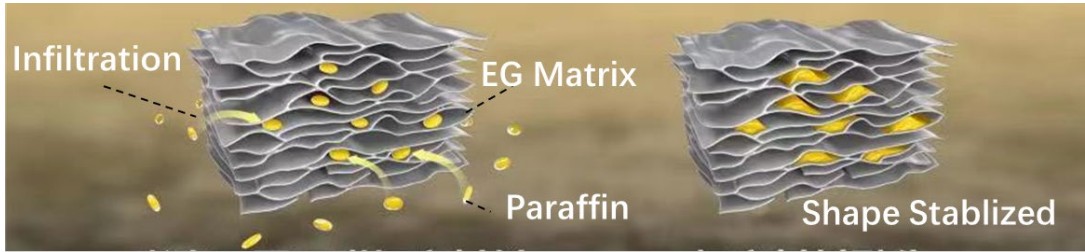

**Figure 3.** Schematic of paraffin infiltrated into the EG matrix.

The maximum loading percentage of paraffin in the EG matrix is affected by the density and the temperature. The adsorption rate of n-tetradecane varying with the time is plotted in Figure 4. Obviously, more paraffin is adsorbed by the EG over time. However, as the density increases, the maximum adsorption rate decreases linearly. The increased density squeezes the pore volume, thus less paraffin can be contained by the porous EG matrix. As the EG density increases by $100\,\text{kg/m}^3$, the loading percentage of paraffin is reduced by 10.2 wt% on average.

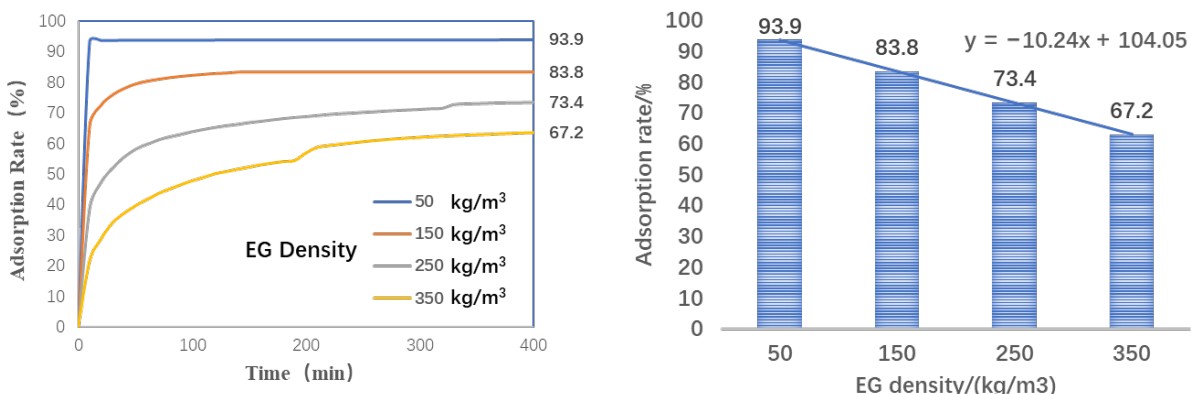

**Figure 4.** Adsorption rate of n-tetradecane in EG with different densities.

We anticipated that a higher temperature could lower the maximum loading percentage of paraffin, as a lower viscosity and surface tension could empower the liquid paraffin to move around more easily. Figure 5 does show that the less paraffin can be adsorbed by the EG matrix. However, the maximum content of paraffin seems to be insensitive to the temperature. As the temperature increases from 15 to 55 °C, the maximum weight ratio of n-tetradecane only decreases from 84.3% to 83.5%.

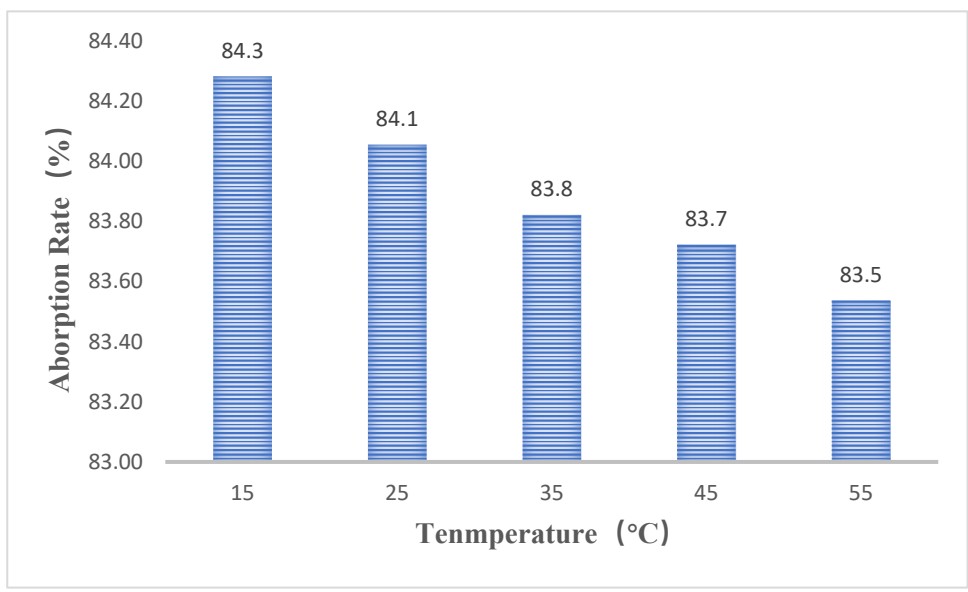

**Figure 5.** Adsorption rate of n-tetradecane in the EG matrix (150 kg/m³) under different temperature.

### 3.2. Thermophysical Properties of n-Tetradecane/EG

The thermal conductivity of the n-tetradecane/EG composite can be enhanced with the EG by up to a hundred times. For the raw paraffin, its thermal conductivity is about 0.2 W/m·K. After it infiltrates into the EG matrix, the thermal conductivity increases up to 21.0 W/m·K. The EG density has a significant impact on the thermal conductivity of the composite PCM. A larger density helps the EG form a denser thermal conduction network but offers a smaller space to contain the paraffin, and the mass percentage of n-tetradecane decreases. Thus, the thermal conductivity increases with the EG density, but the phase change enthalpy decreases. As shown in Figure 6, the thermal conductivity increases linearly with the increase in EG density. On the contrary, the phase change enthalpy drops with a higher EG density, due to the lower adsorption rate of n-tetradecane mentioned in Section 3.1. Table 1 summarizes the melting and solidification enthalpy of the composite PCM. The values measured experimentally agree with the product of the phase change enthalpy of the raw n-tetradecane and its mass percentage in the composite PCM, which are called the theoretical values.

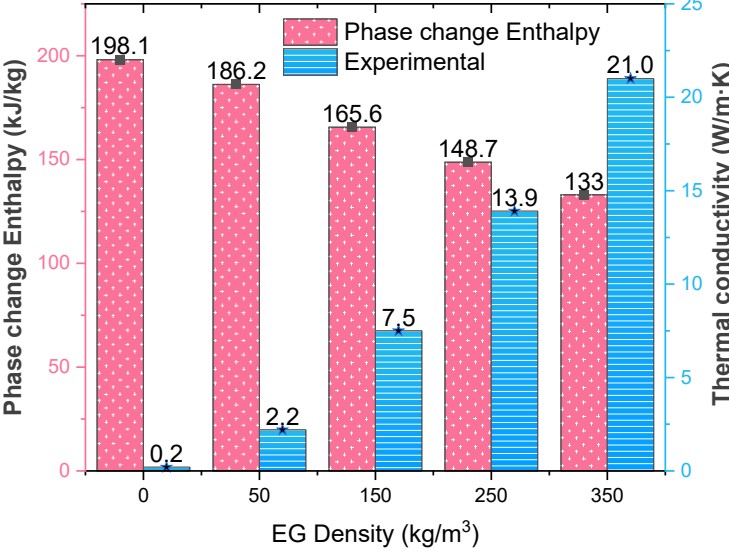

**Figure 6.** Variation of phase change enthalpy and thermal conductivity of n-tetradecane/EG with the EG density.

**Table 1.** Phase change enthalpy of the n-tetradecane/EG with different EG density.

| EG Density (Kg/m$^3$) | Mass Percentage of n-Tetradecane/% | Density of Composite PCM (Kg/m$^3$) | Melting Enthalpy/(J/g) | | Solidification Enthalpy/(J/g) | |
|---|---|---|---|---|---|---|
| | | | Theoretical | Experimental | Theoretical | Experimental |
| 350 | 67.2 | 1067.1 | 138.9 | 133.0 | 135.8 | 132.8 |
| 250 | 73.4 | 939.8 | 145.3 | 142.7 | 145.1 | 143.4 |
| 150 | 83.8 | 925.9 | 170.4 | 165.6 | 167.9 | 165.2 |
| 50 | 93.9 | 819.7 | 193.6 | 186.2 | 189.9 | 185.8 |
| 0 | 100 | 735.0 | 198.1 | 198.1 | 197.7 | 197.7 |

*3.3. Thermophysical Property Model of n-Tetradecane/EG*

Since the phase change enthalpy of the composite PCM is greatly affected by the compositions, it is hard to do experiments to measure the thermophysical properties that cover all the compositions. We attempt to simplify the process of obtaining thermophysical properties with a thermophysical model.

First, to capture the maximum adsorption rate of paraffin in EG, we illustrate a structure of the EG-based composite, as Figure 7 shows. From this figure, we have the following hypothesis: the density of the EG $\rho_{EG}$ is variable, but the density of the paraffin $\rho_P$ remains constant, which is 735 kg/m$^3$ for the n-tetradecane. When the EG is compacted, only the volume of EG is squeezed. The volume of paraffin remains the same. In the sample with a higher density, there is then less paraffin that can be confirmed in the pores of EG. Then the maximum adsorption ratio of paraffin in the EG-based composite PCM $\alpha$ can be calculated with the following equations:

$$\alpha = \frac{\rho_P}{\rho_P + \rho_{EG}} = \frac{\rho_P}{\rho_P + \rho_{PCM}(1 - \alpha)} \tag{3}$$

$$\alpha = \frac{(\rho_{PCM} + \rho_P) - \sqrt{(\rho_G + \rho_P)^2 - 4(\rho_G \rho_P)}}{2\rho_{PCM}} \tag{4}$$

where the density of paraffin $\rho_P$ equals 735 kg/m$^3$, $\rho_{PCM}$ represents the density of the composite PCM. The $\alpha$ can be calculated with different $\rho_{PCM}$. The predictive results have been validated, which are plotted in Figure 8. The maximum adsorption rate can be predicted accurately, with a maximum error of 1.7%. The highly precise adsorption rate allows us to quickly calculate the phase change enthalpy, a byproduct of the adsorption rate and the phase change enthalpy of the raw paraffin.

**Figure 7.** Structure schematic of the composite PCM in different density.

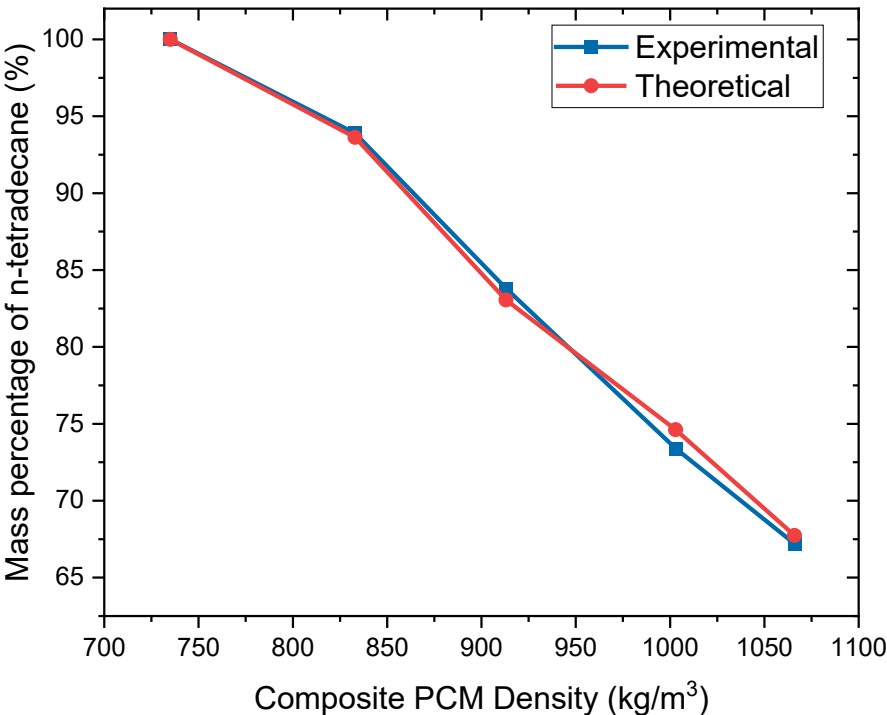

**Figure 8.** Validation of the model to predict the maximum adsorption rate of paraffin.

Apart from the prediction of the thermal storage density, we also develop a simple model to predict the thermal conductivity, which is explained as follows: the contribution of paraffin is ignored, and only EG is considered to enhance the heat transfer. Therefore, we know the thermal conductivity of the composite PCM if the thermal conductivity of EG is calculated. As shown in Figure 9, graphite is a dense carbon material. When it is expanded to EG, voids take place between different layers of the EG skeleton. These voids lower the thermal conductivity of the graphite. We present an effectiveness ratio $\beta$ to characterize the impact of EG on the enhancement of the thermal conductivity of graphite.

$$\beta = \frac{\rho_{EG}}{\rho_G} \tag{5}$$

where $\rho_{EG}$ is the density of the EG and $\rho_G$ is the density of the natural graphite, which equals 2333 kg/m$^3$. Then the thermal conductivity of EG $k_{EG}$ is calculated by the product of $\beta$ and the thermal conductivity of natural graphite $k_G$, which is 129 W/m·K.

$$k_{EG} = \beta \cdot k_G \tag{6}$$

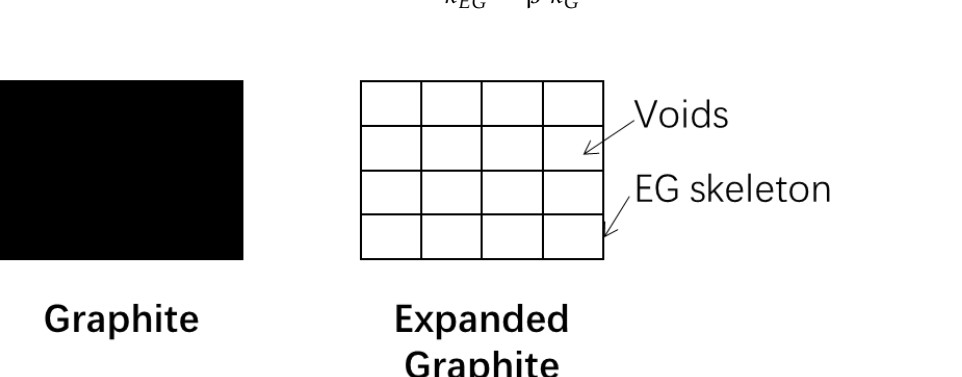

**Figure 9.** Thermal network variation from graphite to expanded graphite (EG).

The thermal conductivity of the composite PCMs with different EG densities has been predicted and compared with the experiment data, as plotted in Figure 10. The thermal conductivity model is also very accurate; the average prediction error is less than 10%.

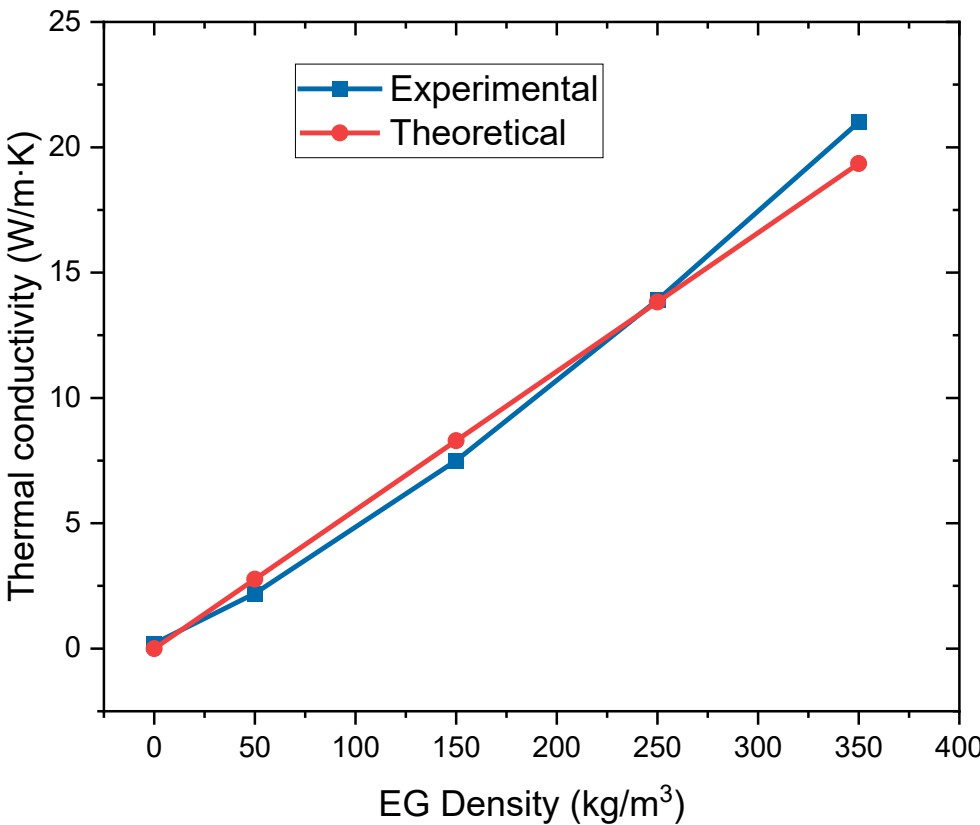

**Figure 10.** Validation of the thermal conductivity model.

The maximum adsorption rate model and the thermal conductivity model have been verified with the experiment data, which could greatly help us save time in measuring the thermophysical properties.

*3.4. Cold Storage Release Performance of the Spiral Wavy Plate Tank*

We study the impacts of thermal conductivity on the discharge performance of a spiral wavy plate cold storage tank, which is filled with the shape-stabilized n-tetradecane/EG composite. Figure 11 shows that a higher thermal conductivity accelerates the cold discharge process. The warm water at 10 °C flows through the cold storage tank to produce chill water with a temperature lower than 9 °C for air conditioning. If we use the raw n-tetradecane, its low thermal conductivity severely limits the cooling efficiency. The outlet temperature of water comes into a plateau around 8.5 °C and the discharge lasts for about 700 s.

The heat transfer efficiency can be significantly enhanced by the composite PCMs with a higher thermal conductivity. When the PCM increases to 2.5 W/m·K, the outlet temperature plateau drops to 6.6 °C and the time for one full discharge has been shortened to 350 s. Once the PCM thermal conductivity increases to 7.5 W/m·K, the discharge time can be reduced to 282 s, while the temperature plateau is lowered down to 6.1 °C. However, the discharge performance is not further improved with the thermal conductivity increasing to between 13.9 and 21.0 W/m·K.

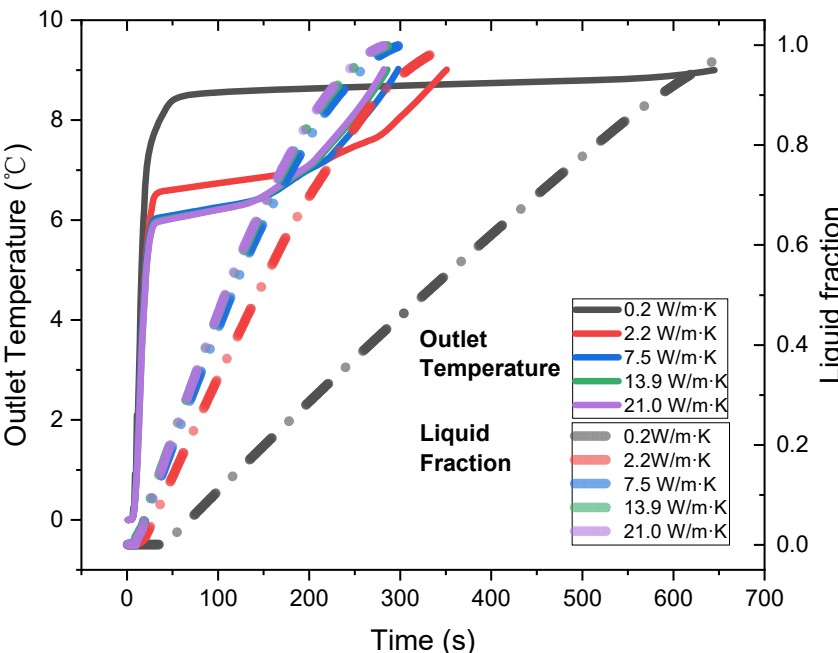

**Figure 11.** Out temperature and liquid fraction of the cold storage tanks filled with PCMs of different thermal conductivities.

Figure 12 compares the power density and energy storage density of the cold storage tank. Obviously, a higher thermal conductivity allows the cold storage tank to release the cold thermal energy faster, and its power density increases monotonously. The energy storage density slightly varies with the thermal conductivity. However, a close look shows that the energy storage density increases from 32.7 to 36.1 kWh/m$^3$ first but then drops to 33.3 kWh/m$^3$, as the thermal conductivity increases. Overall, the energy density of the tank with composite PCMs is higher than the raw n-tetradecane. The results seem not to match with the phase change enthalpy of the materials, which decreases monotonously with the increase in thermal conductivity. However, the results do make sense. The low thermal conductivity limits the cold release capability of the raw n-tetradecane. The discharge ends once the outlet temperature reaches 9 °C. At that point, only 96% of the n-tetradecane has been utilized to release the cold, as shown in the liquid fraction in Figure 11. In comparison, the utilization rate reaches 99.9% for the composite PCM with thermal conductivity enhanced. In the meantime, the mass density of the composite PCMs is 10–27% higher than the raw n-tetradecane, which also compensates for the energy density loss. In summary, the more efficient heat transfer and higher mass density provide the composite PCMs with a higher power and energy density. However, notice that the enhanced thermal conductivity still sacrifices the energy storage density of the materials, which causes the energy density drops from 36.1 to 33.3 kWh/m$^3$.

Figure 13 visually displays the difference in the temperature field and liquid fraction of the raw n-tetradecane and the n-tetradecane/EG composite with a thermal conductivity of 7.5 W/m·K. From the temperature field contours, we can see that heat diffuses more rapidly in the composite PCM. After 300 s, the composite PCMs of the first 3 layers from outside reach 10 °C. However, only part of the raw n-tetradecane near the wall of the wavy plate melts, even after 600 s have passed. For the most part, their temperature is less than 8 °C.

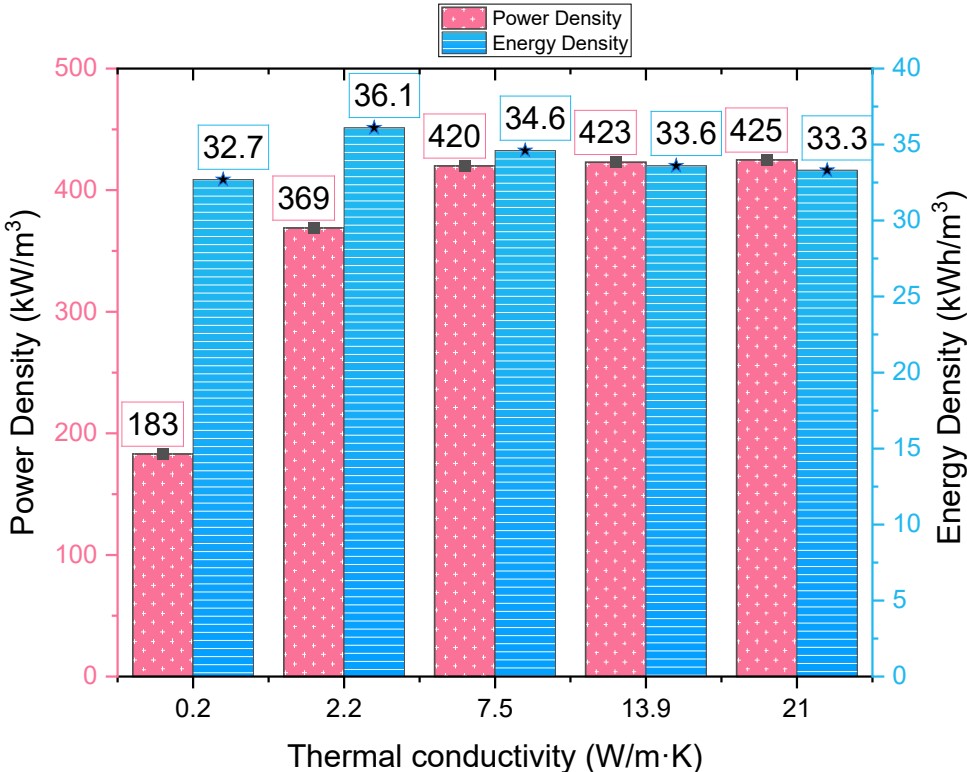

**Figure 12.** Energy and power density with the PCMs of different thermal conductivities.

From the liquid fraction contours, we can see the wavy structure of the spiral plate could enhance the heat transfer between the thermal fluid and the raw n-tetradecane. When the fluid flows to the peaks of the wavy plate, turbulence is enhanced locally. Thus, the n-tetradecane near the peaks melts faster. The local warmer areas can be observed along with the wavy plate periodically. However, the high thermal conductivity ensures a more uniform melting process inside the composite PCM.

From Figure 13, we observe that the composite PCMs melt faster and are more spatially uniform.

At last, the cold storage tank structure is optimized to improve its energy density. Three configurations are shown in Figure 14, in which the PCM thickness per layer differs from 10 to 20 mm. With the PCM thickness per layer increasing, the volume and total heat transfer surface of the spiral wavy plates decreases. The PCM mass remains the same among these three tanks, thus they can store the same energy in total. Figure 14 also compares the temperature field and liquid fractions of the cold storage tanks with different PCM thicknesses. Clearly, we can see that with a thicker layer of PCM, the PCM temperature is lower, which means less cold is released.

Figure 15 compares the outlet temperature and liquid fraction of the PCMs in different configurations. The increase in the PCM thickness takes the cold storage tank more time to release the cold thermal energy and the outlet temperature increases as well. Due to the shrink of the heat transfer area, the heat transfer rate decreases, and the cold release rate is slowed down. Figure 16 shows that as the PCM thickness increases from 10 to 20 mm, the power density decreases by 10.2%. However, the energy density increases by 16.2% even though the PCM weight remains the same. The high thermal conductivity composite PCMs save the volume of heat exchangers. Thus, a greater energy density can be achieved at a smaller cost of power loss.

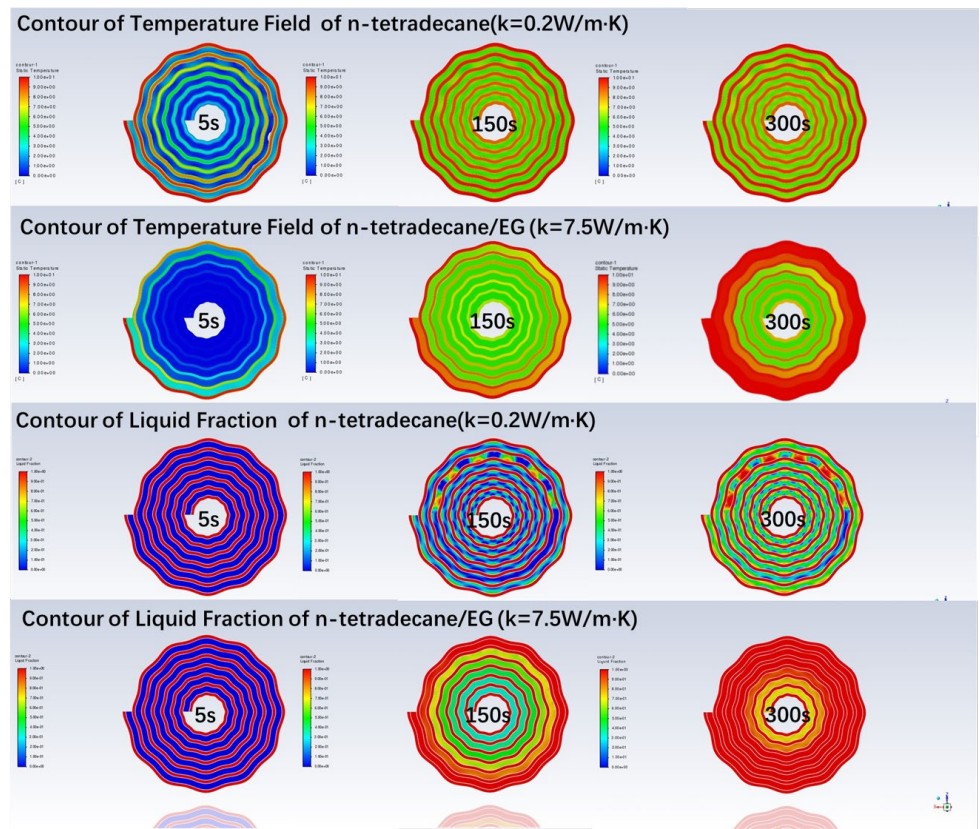

**Figure 13.** Contours of temperature and liquid fraction at different time.

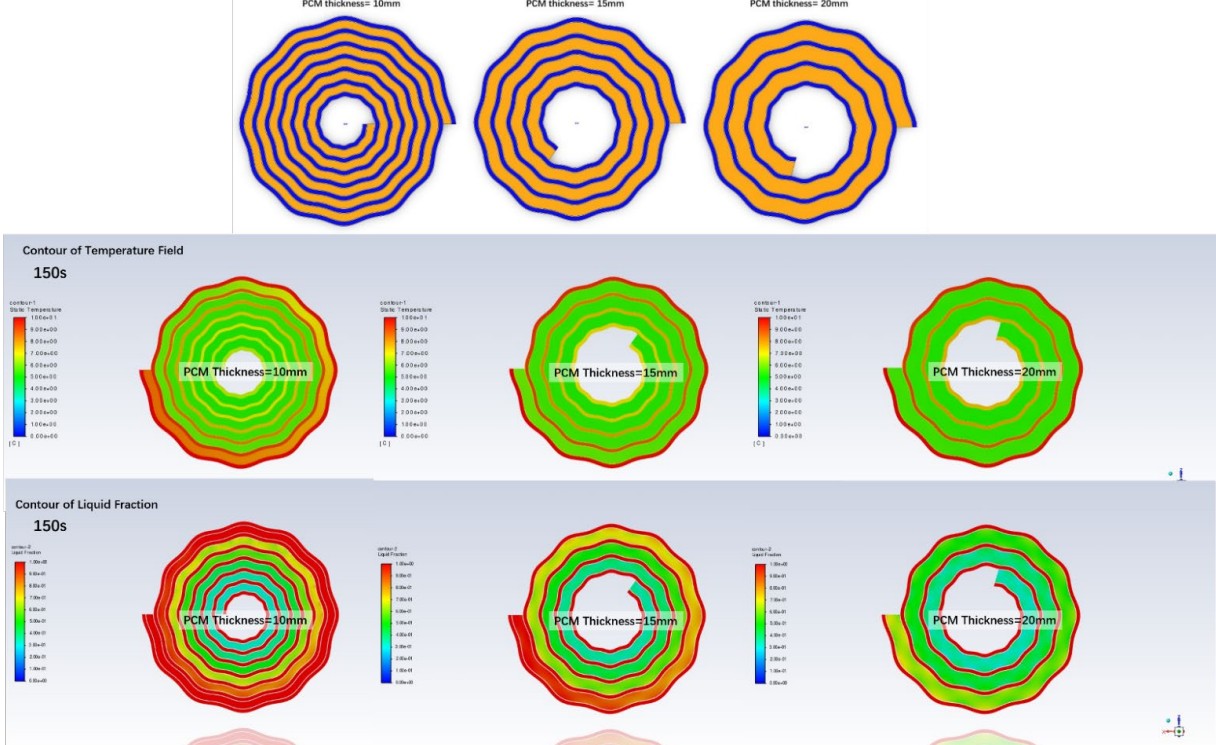

**Figure 14.** Cold storage tanks structure with various PCM thickness per layer, and contours of temperature field and liquid fraction at 150 s during a discharge.

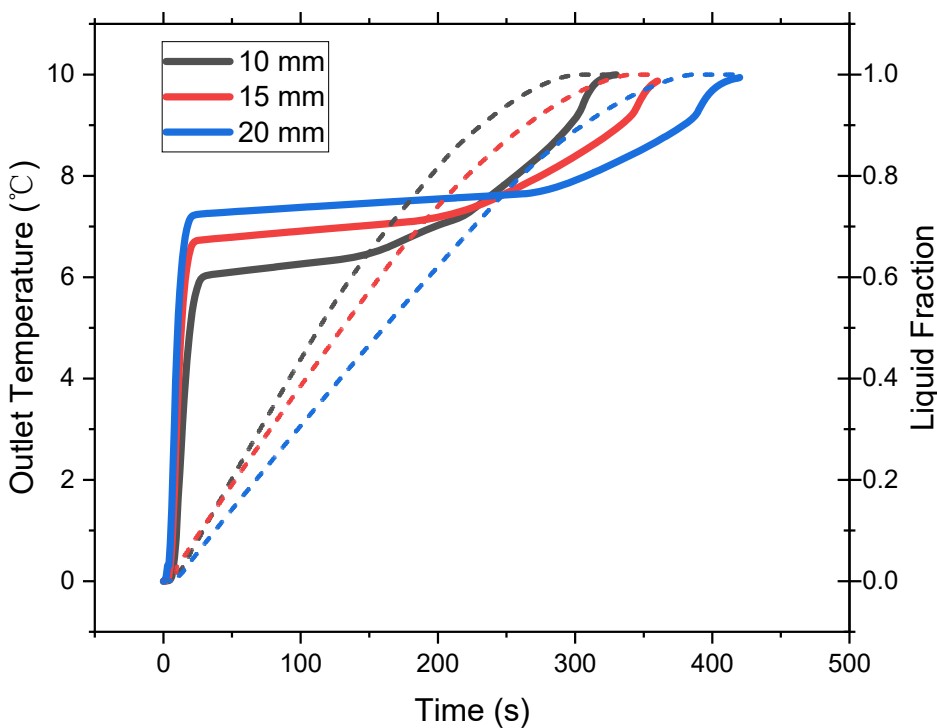

**Figure 15.** Discharge performance of cold storage tanks with different PCM thickness.

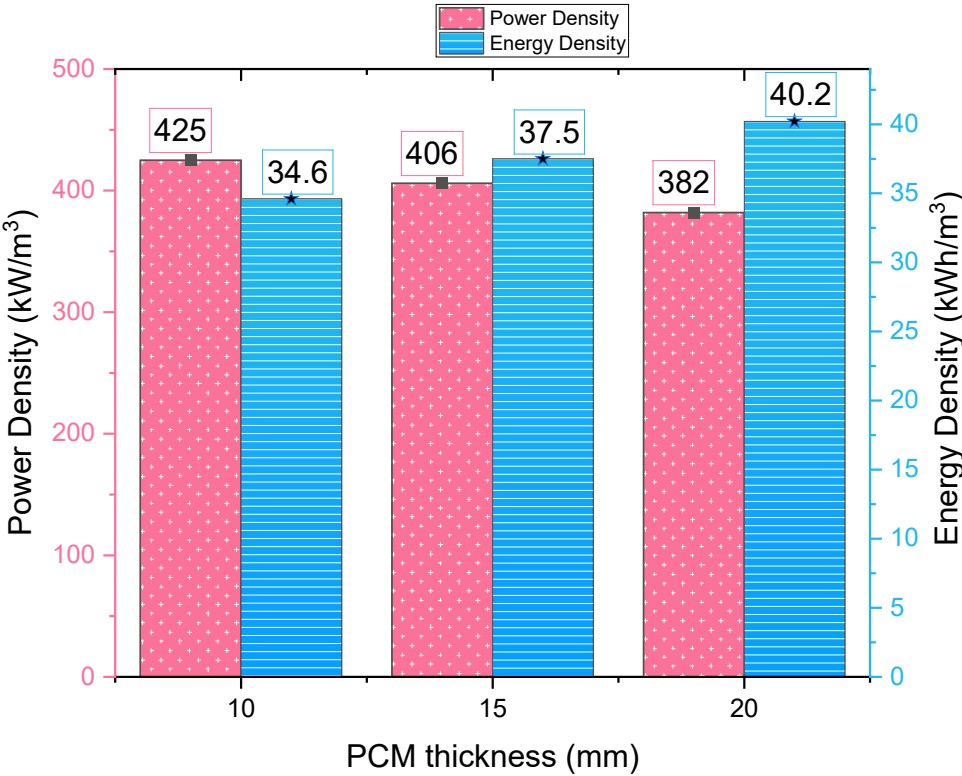

**Figure 16.** Power and energy density of the storage tanks with different PCM thickness.

## 4. Conclusions

This paper prepares an n-tetradecane/EG composite PCM with a high thermal conductivity of up to 21.0 W/m·K. The composite PCM has a suitable melting point of 4 °C and a melting enthalpy of 132.8–189.9 J/g, which are suitable for cold storage in air conditioning systems. One model has been presented to predict the maximum adsorption rate of

n-tetradecane in EG with a maximum error of 1.7%, which helps calculate the maximum phase change enthalpy of the composite PCM under a certain density. Another model is presented for thermal conductivity prediction, the average error of which is within 10%. The two models could give theoretical thermophysical properties of composite PCMs for cold storage tank design. This paper also presents a novel spiral wavy plate tank for cold storage in air conditioning systems. We find that the increase in thermal conductivity of PCMs allows the cold storage tank to have a higher power density and energy storage density in space. The energy density reaches 40.2 kWh/m$^3$, which is high among the organic PCMs. The composite PCM with a higher thermal conductivity is denser in mass and requires less heat transfer area, thus it performs much better than the raw paraffin.

**Author Contributions:** Conceptualization, H.Z. and T.W.; methodology, H.Z., T.W. and L.T.; writing—original draft preparation, H.Z., T.W. and Z.L.; writing—review and editing, Z.L., Z.Z. and X.F.; All authors have read and agreed to the published version of the manuscript.

**Funding:** This work was funded by the Key Science and Technology Projects in Key Areas of Foshan with a grant number of No. 2120001008795.

**Institutional Review Board Statement:** Not applicable.

**Informed Consent Statement:** Not applicable.

**Data Availability Statement:** Not applicable.

**Conflicts of Interest:** The authors declare no conflict of interest.

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
