# Peer review of "Preparation and Thermal Model of Tetradecane/Expanded Graphite and A Spiral Wavy Plate Cold Storage Tank"

_energies, doi:10.3390/en15249435_

Round 1
Reviewer 1 Report
In this manuscript, a low-temperature CPCM was prepared by used n-tetradecane and EG as the PCM and the thermal conductivity enhancer, respectively. The thermal conductivity of the CPCM above 20 W/m·K, which is a promising candidate for low-temperature utilization. In addition, the authors put forward with a novel heat exchanger with a spiral wave plate. It is worthy of recommended and published. List the following suggestions:
1. Page 7, Line 89, “cPCM” should be “CPCM”.
2. Page 3, Line 99, “10℃” should be “10 oC”.
3. The Error message "Error! Reference source not found " appears several times in the manuscript, please re-examine the manuscript.
4. When the material was prepared, graphite impregnated with n-tetradecane was used. Was it prepared under vacuum environment?
Author Response
In this manuscript, a low-temperature CPCM was prepared by used n-tetradecane and EG as the PCM and the thermal conductivity enhancer, respectively. The thermal conductivity of the CPCM above 20 W/m·K, which is a promising candidate for low-temperature utilization. In addition, the authors put forward with a novel heat exchanger with a spiral wave plate. It is worthy of recommended and published. List the following suggestions:
- Page 7, Line 89, “cPCM” should be “CPCM”.
Response: “cPCM” has been corrected as “CPCM”.
- Page 3, Line 99, “10℃” should be “10 oC”.
Response: “10℃” has been corrected as “10 °C”.
- The Error message "Error! Reference source not found " appears several times in the manuscript, please re-examine the manuscript.
Response: All the figure captions have been corrected.
- When the material was prepared, graphite impregnated with n-tetradecane was used. Was it prepared under vacuum environment?
Response: It was prepared under normal atmosphere. No vacuum environment is needed.
Reviewer 2 Report
The paper presents the preparation of a composite phase change material, CPCM, using tetradecane and expanded graphite. Two models, used to estimate the density and thermal conductivity of the CPCM, were developed and validated with experimental results. Additionally, a spiral wave plate cold storage tank was designed to use with de CPCM. The simulation results show a significant improvement of power and energy density, in comparison with the tetradecane. I would recommend it for publication once the following minor concerns have been addressed:
1. I recommend a thorough revision of the manuscript, tables and figures. There are several minor mistakes, for example, in Table 1, line 163, the column for the density of CPCM is void.
2. Please provide more information about the criteria for the design of the wavy structure of the cold storage unit, lines 93-99. Moreover, I recommend to include the basic geometrical parameters.
3. Lines 136-141, please elaborate about why the maximum content of paraffin seems to be unsensitive to the temperature.
4. The contribution of the presented research should be clearly stated both in the abstract and the conclusions.
Author Response
The paper presents the preparation of a composite phase change material, CPCM, using tetradecane and expanded graphite. Two models, used to estimate the density and thermal conductivity of the CPCM, were developed and validated with experimental results. Additionally, a spiral wave plate cold storage tank was designed to use with de CPCM. The simulation results show a significant improvement of power and energy density, in comparison with the tetradecane. I would recommend it for publication once the following minor concerns have been addressed:
- I recommend a thorough revision of the manuscript, tables and figures. There are several minor mistakes, for example, in Table 1, line 163, the column for the density of CPCM is void.
Response: Based on your and other reviewers’ comments, tables and figures have been checked and revised. The void column has been filled.
- Please provide more information about the criteria for the design of the wavy structure of the cold storage unit, lines 93-99. Moreover, I recommend to include the basic geometrical parameters.
Response: The criteria for the design of the cold storage unit has been added. The govern equations of the spiral wavy structure have been added.
“Its energy density was supposed to be 200 kJ, with a discharge power of 0.5 kW. This cold storage tank is one tenth unit of a cold storage system with a capacity of 0.5 kWh and 5 kW, which is used for emergency cooling for a small power system”
The wavy boundary of the spiral plates were governed by the following equations:
- Lines 136-141, please elaborate about why the maximum content of paraffin seems to be unsensitive to the temperature.
Response: A higher temperature could lower the surface tension of the paraffin, which makes it easier to leak out. But the drop of surface tension only drops slightly. As a result, the maximum content of paraffin seems to be unsensitive to the temperature.
- The contribution of the presented research should be clearly stated both in the abstract and the conclusions.
Response: The abstract and conclusions have been revised.

Reviewer 3 Report
This is a very interesting research paper, which is well-produced. and well presented.
I assume the title of "Figure 11. Out temperature and liquid fraction of the cold storage tanks filled with PCMs of different thermal conductivities."
is
Figure 11. Outlet temperature and liquid fraction of the cold storage tanks filled with PCMs of different thermal conductivities.
In Figure 13. Contours of temperature and liquid fraction at different times should be presented at similar time intervals for comparison reasons.
Temperature contours for different PCM thicknesses will also be useful.
The conclusions are rather general and focused on the models used to assess the material. I believe further statements would improve the conclusions including, energy density compared to other PCM materials
Author Response
Reviewer 3
This is a very interesting research paper, which is well-produced. and well presented.
I assume the title of "Figure 11. Out temperature and liquid fraction of the cold storage tanks filled with PCMs of different thermal conductivities."is Figure 11. Outlet temperature and liquid fraction of the cold storage tanks filled with PCMs of different thermal conductivities.
Response: The extra spacing has been deleted.
In Figure 13. Contours of temperature and liquid fraction at different times should be presented at similar time intervals for comparison reasons.
Response: The intervals have been made the same. At the time of 5s, 150s and 300s, we compare the discharge performance of the cold storage tank with different PCMs.
Temperature contours for different PCM thicknesses will also be useful.
Response: Temperature contours for different PCM thicknesses have been added in Fig. 14.
The conclusions are rather general and focused on the models used to assess the material. I believe further statements would improve the conclusions including, energy density compared to other PCM materials
Response: The conclusions have been revised as “This paper prepares a n-tetradecane/EG composite PCM with a high thermal conductivity up to 21.0 W/m·K. The composite PCM has a suitable melting point of 4 °C and a melting enthalpy of 132.8 ~189.9 J/g, which are suitable for cold storage in air conditioning systems. One model has been presented to predict the maximum adsorption rate of n-tetradecane in EG with a maximum error of 1.7%, which helps calculate the maximum phase change enthalpy of the composite PCM under a certain density. Another model is presented for thermal conductivity prediction, the average error of which is within 10%. The two models could give theoretical thermophysical properties of composite PCMs for cold storage tank design. This paper also presents a novel spiral wavy plate tank for cold storage in air conditioning systems. We find that the increase of thermal conductivity of PCMs allows the cold storage tank to have a higher power density and energy storage density in space. The energy density reaches 40.2 kWh/m3, which is high among the organic PCMs. The composite PCM with a higher thermal conductivity is denser in mass and requires less heat transfer area, thus it performs much better than the raw paraffin. “
The cold storage tank has a relatively high energy density among various organic PCMs thermal storage tanks.